# Dissecting G_q/11_-Mediated Plasma Membrane Translocation of Sphingosine Kinase-1

**DOI:** 10.3390/cells9102201

**Published:** 2020-09-29

**Authors:** Kira Vanessa Blankenbach, Ralf Frederik Claas, Natalie Judith Aster, Anna Katharina Spohner, Sandra Trautmann, Nerea Ferreirós, Justin L. Black, John J. G. Tesmer, Stefan Offermanns, Thomas Wieland, Dagmar Meyer zu Heringdorf

**Affiliations:** 1Institut für Allgemeine Pharmakologie und Toxikologie, Universitätsklinikum, Goethe-Universität, Theodor-Stern-Kai 7, 60590 Frankfurt am Main, Germany; kira.blankenbach@web.de (K.V.B.); frederik.claas@gmx.de (R.F.C.); s5599083@stud.uni-frankfurt.de (N.J.A.); spohner@em.uni-frankfurt.de (A.K.S.); 2Institut für Klinische Pharmakologie, Universitätsklinikum, Goethe-Universität, Theodor-Stern-Kai 7, 60590 Frankfurt am Main, Germany; trautmann@med.uni-frankfurt.de (S.T.); ferreirosbouzas@em.uni-frankfurt.de (N.F.); 3Department of Biochemistry and Biophysics, University of North Carolina, Chapel Hill, NC 27599, USA; justinblack8@gmail.com; 4Departments of Biological Sciences and of Medicinal Chemistry and Molecular Pharmacology, Purdue University West Lafayette, West Lafayette, IN 47907-2054, USA; jtesmer@purdue.edu; 5Abteilung für Pharmakologie, Max-Planck-Institut für Herz- und Lungenforschung, 61231 Bad Nauheim, Germany; offermanns.pharmacology@mpi-bn.mpg.de; 6Experimentelle Pharmakologie Mannheim, European Center for Angioscience, Universität Heidelberg, 68167 Mannheim, Germany; thomas.wieland@medma.uni-heidelberg.de

**Keywords:** sphingosine kinase, sphingosine-1-phosphate, G-protein-coupled receptors, Gα_q/11_

## Abstract

Diverse extracellular signals induce plasma membrane translocation of sphingosine kinase-1 (SphK1), thereby enabling inside-out signaling of sphingosine-1-phosphate. We have shown before that G_q_-coupled receptors and constitutively active Gα_q/11_ specifically induced a rapid and long-lasting SphK1 translocation, independently of canonical G_q_/phospholipase C (PLC) signaling. Here, we further characterized G_q/11_ regulation of SphK1. SphK1 translocation by the M_3_ receptor in HEK-293 cells was delayed by expression of catalytically inactive G-protein-coupled receptor kinase-2, p63Rho guanine nucleotide exchange factor (p63RhoGEF), and catalytically inactive PLCβ_3_, but accelerated by wild-type PLCβ_3_ and the PLCδ PH domain. Both wild-type SphK1 and catalytically inactive SphK1-G82D reduced M_3_ receptor-stimulated inositol phosphate production, suggesting competition at Gα_q_. Embryonic fibroblasts from Gα_q/11_ double-deficient mice were used to show that amino acids W263 and T257 of Gα_q_, which interact directly with PLCβ_3_ and p63RhoGEF, were important for bradykinin B_2_ receptor-induced SphK1 translocation. Finally, an AIXXPL motif was identified in vertebrate SphK1 (positions 100–105 in human SphK1a), which resembles the Gα_q_ binding motif, ALXXPI, in PLCβ and p63RhoGEF. After M_3_ receptor stimulation, SphK1-A100E-I101E and SphK1-P104A-L105A translocated in only 25% and 56% of cells, respectively, and translocation efficiency was significantly reduced. The data suggest that both the AIXXPL motif and currently unknown consequences of PLCβ/PLCδ(PH) expression are important for regulation of SphK1 by G_q/11_.

## 1. Introduction

Sphingosine-1-phosphate (S1P) is a multifunctional lipid mediator involved in organismal development and homeostasis of the immune, cardiovascular, nervous, and metabolic systems [1]. The metabolism of S1P is evolutionarily highly conserved, comprising sphingosine kinases (SphK), lipid phosphate phosphatases, S1P phosphatases, and S1P lyase [2]. In vertebrates, S1P activates five G-protein-coupled receptors (S1P-GPCRs), S1P_1–5_ [3]. These receptors differentially couple to G_i_, G_q/11_, and G_12/13_ proteins, and thereby regulate cell proliferation, survival, migration, adhesion, and Ca^2+^-dependent functions [1]. S1P-GPCRs are ubiquitously expressed and implicated for example in angiogenesis, maintenance of vascular tone and permeability, and immune cell trafficking. Accordingly, S1P-GPCRs play a role in autoimmunity, inflammation, fibrosis, and cancer [1]. Beyond these well-established effects of extracellular S1P, several roles and targets have been described for intracellular S1P. Examples include endocytic membrane trafficking [4,5], Ca^2+^ mobilization, regulation of histone deacetylases, and mitochondrial respiration (reviewed in [6]). Of note, all of these studies show highly localized signaling by SphK, supporting the early conclusion that SphK localization is a key to function [7]. There are two SphK isoforms, which are derived from different genes and differ in tissue expression, structure, subcellular localization, regulation, and function. SphK2 has been observed in the cytosol, ER, mitochondria, and nucleus, whereas SphK1 is mainly found in the cytosol and may translocate to the plasma membrane upon stimulation [7,8,9,10]. Thus, SphK1 seems to be poised to generate S1P for cellular export, and thereby trigger S1P-GPCR cross-activation, which is known as inside-out signaling. A prominent example for inside-out signaling by S1P is observed in fibroblasts, in which SphK1 activation by platelet-derived growth factor (PDGF) leads to cross-activation of the S1P_1_ receptor, which then mediates PDGF-induced cell migration [11]. Numerous other studies have shown the importance of the SphK/S1P axis for auto- and paracrine S1P signaling, and diverse transporters mediating S1P export have been identified (reviewed in [1,12]).

SphK1 is regulated transcriptionally, translationally, and post-translationally by many different pathways [8,9,10]. Acute activation of SphK1, with or without membrane translocation, can be induced by growth factors (for example PDGF, epidermal growth factor, nerve growth factor), cytokines (for example tumor necrosis factor-α, interleukin-1β, transforming growth factor-β), immunoglobulin receptors, or GPCRs (for review, see [8,9,10]). Several mechanisms have been described for plasma membrane translocation of SphK1. Phorbol-12-myristate-13-acetate (PMA)-induced translocation [13] involved phosphorylation of SphK1 at S225 by extracellular signal-regulated kinases (ERK)-1/2 [14]. In contrast, SphK1 translocation by oncogenic K-Ras was dependent on ERK but independent of S225 phosphorylation [15]. SphK1 translocation by both PMA and oncogenic Ras required calcium-and-integrin-binding-protein-1 (CIB1) [16]. Another pathway for acute activation and translocation of SphK1 is phosphatidic acid production [17]. Membrane binding of SphK1 has been attributed to a hydrophobic patch involving L194, F197, and L198 [4]. Furthermore, a highly positively charged site composed of K27, K29, and R186 was shown to form a single contiguous interface with the hydrophobic patch, mediating electrostatic interactions of SphK1 with membranes [18]. Finally, based on SphK1 crystal structures [19,20], Adams et al. have suggested that a dimeric quaternary structure may play a role in curvature-dependent targeting of SphK1 to the plasma membrane, and suggested how phosphorylation at S225 and protein binding to the C-terminus may potentially unmask membrane association determinants in SphK1 [21].

Our own studies have focused on regulation of SphK by GPCRs. Whereas overall SphK activity can be stimulated via G_i_ as well as via G_q_ pathways (reviewed in [22]), we have shown that specifically G_q_-coupled receptors induce a rapid and long-lasting translocation of SphK1 to the plasma membrane [23,24]. SphK1 translocation was further induced by overexpression of constitutively active Gα_q_ and Gα_11_, but not Gα_i_, Gα_12_, or Gα_13_ [23]. Importantly, G_q_-mediated SphK1 translocation was independent of phosphorylation at S225, because SphK1-S225A translocated after stimulation of the M_3_ receptor in HEK-293 cells or the B_2_ receptor in C2C12 myoblasts, similarly to the wild-type enzyme [23,25]. Classical G_q/11_/phospholipase C (PLC) signaling pathways were not involved in SphK1 targeting. Thus, neither cell-permeable diacylglycerol analogues or PMA, which induce activation of protein kinase C, nor thapsigargin or ionomycin, which induce increases in [Ca^2+^]_i_, were able to induce SphK1 translocation to the extent that it was induced by M_3_ receptor stimulation. Even a combined pretreatment with PMA plus ionomycin for about 8 min, which caused a minor SphK1 translocation by itself, did not prevent a subsequent marked SphK1 translocation stimulated by the M_3_ receptor. Furthermore, the involvement of Ca^2+^/calmodulin, phospholipase D, tyrosine kinases, Rho kinase, and mitogen-activated protein kinase kinase was ruled out by specific inhibitors [23].

We, therefore, studied the regulation of SphK1 by G_q/11_ signaling pathways in more detail. We identified and characterized a motif conserved in vertebrate SphK1, with similarities to Gα_q/11_ binding motifs in direct G_q_ effectors, which is required for G_q_-mediated SphK1 translocation. We also showed that PLCβ, beyond its canonical downstream effectors, is important in inducing the most rapid membrane SphK1 translocation upon GPCR stimulation, and that this is mimicked by the pleckstrin homology (PH) domain of PLCδ_1_.

## 2. Materials and Methods

### 2.1. Materials

Carbachol, bradykinin, and fatty-acid-free bovine serum albumin (BSA) were purchased from Sigma-Aldrich (Sigma-Aldrich Chemie GmbH, Taufkirchen, Germany). S1P was from Biomol GmbH (Hamburg, Germany). All other materials were from previously described sources [24,26].

### 2.2. Plasmids

The 3xHA-S1P_1_ in pcDNA3.1 was obtained from the Missouri S&T cDNA Resource Center (Rolla, MO, USA). Gα_qi5_-G66D was kindly provided by Dr. Evi Kostenis (University of Bonn, Bonn, Germany) [27]. The plasmid for expression of Gα_q_-YFP was a kind gift from Dr. Catherine Berlot (Weis Center for Research, Danville, PA, USA) [28]. Plasmids for expression of Gα_i2_, Gα_q_ wild-type, Gα_q_-Q209L-EE, Gα_q_-T257E, Gα_q_-Y261N, Gα_q_-W263D, Gα_q_-D321A, Gα_q_-Y356K, Gα_15_-Q212L-EE, G-protein-coupled-receptor-kinase-2 (GRK2)-K220R, and the bradykinin B_2_ receptor have been described previously [23,29,30,31,32,33].

Plasmids for expression of murine YFP-SphK1 (YFP-mSphK1), human GFP-SphK1 (GFP-hSphK1), human SphK1-G82D (hSphK1-G82D), and human mCherry-SphK1 (mCherry-hSphK1) have been described before [23,24]. Human SphK1-cerulean (hSphK1-cerulean) is a synthetic sequence with optimized codons (Mr. Gene, Regensburg, Germany) deduced from human SphK1 (GenBank accession number AF200328.1) and cerulean fluorescent protein (GenBank accession number ACO48272.1), which was cloned into the pcDNA3.1 vector using HindIII and XhoI. SphK1-F197A-L198Q-GFP and SphK1-L194Q-GFP were kindly provided by Dr. Pietro De Camilli (Yale University School of Medicine, New Haven, CT, USA) [4]. For experiments with the SphK1 mutants, mCherry-hSphK1-A100E-I101E and mCherry-hSphK1-P104A-L105A, these mutants and a second construct of mCherry-hSphK1 wild-type were designed according to the human SphK1 sequence described in GenBank accession number NM_001142601.2. All three constructs were in pmCherry-C1 vector (Clontech/Takara Bio Europe, Saint-Germain-en-Laye, France) and obtained from Proteogenix (Schiltigheim, France). Full-length p63Rho guanine nucleotide exchange factor (p63RhoGEF) in pmCherry-C1 vector was a kind gift from Dr. Dorus Gadella (University of Amsterdam, Amsterdam, The Netherlands; Addgene plasmid #67896; http://n2t.net/addgene:67896; RRID:Addgene_67896) [34]. PLCβ_3_ (GenBank accession number NM_000932), C-terminally tagged with TurboGFP in pCMV6-AC-GFP vector, was obtained from OriGene Technologies (product #RG224268; Rockville, MD 20850, USA). PLCβ_3_-H332A-GFP in pCMV6-AC-GFP vector was obtained from Proteogenix (Schiltigheim, France). PLCδ_1_(PH)-CFP was a kind gift from Dr. Michael Schäfer (University of Leipzig, Leipzig, Germany) [35].

### 2.3. Cell Culture and Transfection

HEK-293 cells stably expressing the M_3_ muscarinic acetylcholine receptor were cultured in Dulbecco’s modified Eagle’s medium (DMEM/F12) supplemented with 10% fetal calf serum, 100 U/mL penicillin G, and 0.1 mg/mL streptomycin as described [23]. Stock cultures of HEK-293 cells were grown in the presence of 0.5 mg/mL G418. Mouse embryonic fibroblasts (MEFs) from CIB1-deficient mice were made by J.L. Black in the laboratory of Dr. Leslie V. Parise (University of North Carolina at Chapel Hill, NC, USA) [36,37]. These MEFs, along with MEFs from Gα_q/11_ double-deficient mice [38], were cultured in DMEM/F12 medium with 10% fetal calf serum, 100 U/mL penicillin G, and 0.1 mg/mL streptomycin. Transfection of HEK-293 cells was performed with Lipofectamine 2000 (Invitrogen GmbH, Karlsruhe, Germany), while MEFs were transfected with Turbofect (Fermentas, St. Leon-Rot, Germany) according to the manufacturer’s instructions. For microscopy, the cells were seeded onto poly-l-lysine-coated 8-well slides (μ-slide; ibidi GmbH, Martinsried, Germany). Before experiments, the cells were kept in serum-free medium overnight.

### 2.4. Measurement of SphK1 Translocation

SphK1 translocation was analyzed using fluorescently labelled SphK1 constructs and confocal laser scanning microscopy as described recently [24]. Cells grown on poly-l-lysine-treated 8-well slides (μ-slide; ibidi GmbH, Martinsried, Germany) were incubated in Hank´s balanced salt solution (HBSS) containing 118 mM NaCl, 5 mM KCl, 1 mM CaCl_2_, 1 mM MgCl_2_, 5 mM glucose, and 15 mM HEPES, pH 7.4. Fluorescence microscopy was performed with a Zeiss LSM510 Meta inverted confocal laser scanning microscope equipped with a Plan-Apochromat 63×/1.4 oil immersion objective (Carl Zeiss MicroImaging GmbH, Göttingen, Germany). The following excitation (ex) laser lines and emission (em) filter sets were used: CFP and cerulean: ex 458 nm, em band-pass 465–510 nm; GFP: ex 488 nm, em long-pass 505 nm; GFP in combination with mCherry: ex 488 nm, em band-pass 505–530; YFP: ex 514, em band pass 525–600; mCherry: ex 561 nm, em band-pass 575–630 nm. Translocation half-times were determined by measuring the fluorescence intensity within defined cytosolic regions and fitting exponential functions to the translocation-induced decay in cytosolic fluorescence intensity. For estimation of translocation efficiency, the translocated fraction was calculated from these exponential curves as % decay in cytosolic fluorescence. While the average translocation half-time was ~5 s throughout all experiments, the translocated fraction values were comparatively variable between experiments because their calculation was influenced to a certain degree by cell shape changes. For quantification of SphK1 plasma membrane localization in cells co-transfected with Gα_q_-Q209L-EE (Figure 6G,H), we measured the fluorescence profiles of individual cells using the ZEN software (Carl Zeiss MicroImaging GmbH, Göttingen, Germany), and calculated the ratios of plasma membrane and cytosolic fluorescence intensities.

### 2.5. Inositol Phosphate Production

Inositol phosphate production was measured as described recently [24]. Briefly, HEK-293 cells labelled with 1 µCi/mL myo-2-[^3^H]-inositol (23.75 Ci/mmol; Perkin Elmer Life and Analytical Sciences, Rodgau-Jügesheim, Germany) were stimulated with 100 µM carbachol in HBSS containing LiCl for 20 min at 37 °C. The reaction was stopped by addition of 2 mL ice-cold methanol. The cells were scraped from the dishes, 1 mL H_2_O and 2 mL chloroform were added, and the aqueous phase was transferred to Poly-Prep AG 1-X8 columns (Bio-Rad, Hercules, CA, USA). After washing with H_2_O and 50 mM ammonium formate, inositol phosphates were eluted with 5 mL of 1 M ammonium formate and 0.1 M formic acid. The radioactivity was measured by liquid scintillation counting.

### 2.6. Western Blotting

Cells grown to near confluence on 6 cm-dishes were lysed, the proteins were separated by SDS gel electrophoresis and blotted onto polyvinylidene difluoride membranes. The SphK1 antibody, directed against the C-terminus of human SphK1, was a kind gift from Drs. Andrea Huwiler (University of Bern, Bern, Switzerland) and Josef Pfeilschifter (Goethe-University Frankfurt, Frankfurt, Germany) [39]. Anti-mCherry antibody (ab125096) was from Abcam (Cambridge, UK). Anti-β-actin (A5441) was from Sigma-Aldrich Chemie GmbH (Taufkirchen, Germany), HRP-conjugated secondary antibodies were from GE Healthcare (Freiburg, Germany), and the enhanced chemiluminescence system was from Millipore Corporation (Billerica, MA, USA).

### 2.7. High-Performance Liquid Chromatography Tandem Mass Spectrometry

S1P (d18:1) concentrations were determined by high-performance liquid chromatography tandem mass spectrometry as described recently [24].

### 2.8. Data Analysis and Presentation

Fluorescence images were edited with the ZEN software (Carl Zeiss MicroImaging GmbH, Göttingen, Germany). Statistical tests, curve fitting, and calculations of translocation half-times were done with Prism-5 (GraphPad Software, San Diego, California, USA). Averaged data are expressed as means ± SD or means ± SEM from the indicated number (*n*) of cells, samples, or experiments, respectively.

## 3. Results and Discussion

Similar to our previous publications [23,25], we observed a rapid and long-lasting translocation of both murine and human SphK1 to the plasma membrane upon stimulation of the M_3_ muscarinic acetylcholine receptor in HEK-293 cells (Figures 1, 2, 5, and 6). To confirm the involvement of Gα_q/11_ in this system, we analyzed the influence of a catalytically inactive mutant of GRK2, GRK2-K220R. GRK2-K220R is unable to phosphorylate G protein-coupled receptors but directly binds Gα_q/11_ and Gβγ, and acts as a Gα_q/11_/Gβγ scavenger [29] (see also [40]). When co-expressed with YFP-mSphK1, GRK2-K220R strongly delayed, reduced, and in several cells even fully prevented translocation of the enzyme by the M_3_ receptor (Figure 1A–C). Together with our previous observation that constitutively active Gα_q_ induced SphK1 translocation, this result indicated that M_3_ receptor-induced translocation was mediated by Gα_q_.

Another binding partner and direct effector of Gα_q/11_ is p63RhoGEF [41,42]. It is known that p63RhoGEF activates RhoA but also competes with PLCβ for activated Gα_q/11_, and vice versa [41]. Therefore, we tested the influence of p63RhoGEF on SphK1 translocation. As shown in Figure 1D–F, overexpressed mCherry-p63RhoGEF strongly delayed and reduced M_3_ receptor-induced translocation of GFP-hSphK1. This result indicated that SphK1 is not activated by p63RhoGEF downstream signaling and matches our previous observation that a Rho kinase inhibitor did not prevent G_q//11_–mediated SphK1 translocation [23]. The data obtained with GRK2-K220R and p63RhoGEF can be explained by competition of the different Gα_q/11_ effectors for binding at the active Gα_q_ or Gα_11_ subunit. Thus, the data suggest two possibilities: (1) that SphK1 is activated downstream of PLCβ, and (2) that SphK1 directly interacts with and is translocated by active Gα_q/11_. To analyze these possibilities further, we next tested the influence of PLCβ on SphK1 translocation.

As shown in Figure 2A–D, overexpression of catalytically inactive PLCβ_3_-H332A-GFP caused a significant delay in M_3_ receptor-induced translocation of mCherry-hSphK1. In contrast, wild-type PLCβ_3_-GFP slightly but significantly accelerated translocation of mCherry-hSphK1. While the inhibitory effect of PLCβ_3_-H332A could be caused both by competition at Gα_q_ and by its activity as a GTPase-activating protein (GAP), the acceleration by wild-type PLCβ_3_ shows that its GAP activity was surmounted by its stimulatory effect. Interestingly, acceleration of M_3_ receptor-induced SphK1 translocation was also observed upon expression of the PH domain of PLCδ_1_, which serves as a sensor for phosphatidylinositol-4,5-bisphosphate (PI(4,5)P_2_) [43]. As described [35,43], PLCδ_1_(PH)-CFP was localized at the plasma membrane in unstimulated cells and rapidly translocated to the cytosol after stimulation of the M_3_ receptor (Figure 2E). The velocity of PLCδ_1_(PH)-CFP translocation was not altered by co-expression of YFP-mSphK1; however, translocation of YFP-mSphK1 was significantly accelerated by co-expression of PLCδ_1_(PH)-CFP (Figure 2F). Since the PH domain of PLCδ_1_ does not induce DAG and IP_3_ production, its effect on SphK1 is rather due to PI(4,5)P_2_ binding or other cellular effects, for example competition with other PI(4,5)P_2_ binding proteins. In fact, PLCδ_1_(PH) has been shown to reduce the amount of phosphatidylinositol-4-phosphate-5-kinase (PIP5K) at the plasma membrane, and thereby the cellular level of PI(4,5)P_2_ [44]. According to this report, expression of PLCδ_1_(PH) will rather decrease than increase PLCβ catalytic activity, but nevertheless accelerated SphK1 translocation. Importantly, RhoA, the downstream effector of p63RhoGEF, activates type I PIP5K [45]. Thus, it is possible that (over)expression of PLCβ_3_ or PLCδ_1_(PH) accelerated SphK1 translocation while p63RhoGEF delayed SphK1 translocation by decreasing and enhancing PI(4,5)P_2_ levels, respectively. However, other tools manipulating PI(4,5)P_2_ levels such as the multiple pathway inhibitor genistein [46] did not alter SphK1 translocation velocity in our cells ([23] and data not shown). Consequently, the mechanisms by which (over)expression of PLCβ_3_ and PLCδ_1_(PH) accelerate SphK1 translocation remain unclear.

Interestingly, not all members of the Gα_q_ subfamily (Gα_q_, Gα_11_, Gα_14_, and Gα_15/16_, with Gα_15_ being the murine homologue of human Gα_16_) interact with the established targets in the same manner. For example, Gα_15/16_ does not bind to GRK2, and binds to but does not activate p63RhoGEF (reviewed in [40]). With the aim of possibly separating PLCβ activation and SphK1 translocation, we expressed constitutively active Gα_15_-Q212L-EE and studied its influence on SphK1 localization. As shown in Figure 3A, mCherry-SphK1 was strongly localized at the plasma membrane in Gα_15_-Q212L-EE-transfected cells. Thus, PLCβ activation and SphK1 translocation could not be separately targeted by using Gα_15/16_.

To further analyze the mutual interactions of SphK1 and PLCβ, we next studied the influence of overexpressed SphK1 on M_3_ receptor-induced accumulation of [^3^H]inositol phosphates in [^3^H]inositol-labelled cells. As shown in Figure 3B, GFP-SphK1 had no influence on basal inositol phosphate production, but significantly reduced M_3_ receptor-stimulated inositol phosphate production. Moreover, SphK1-G82D, which is a catalytically inactive mutant [47], had the same effect (Figure 3C), indicating that it was due to protein–protein interactions and independent of S1P signaling. This result indeed suggests that SphK1 competed with PLCβ for Gα_q/11_ in the context of a living cell, although it remains possible that SphK1 binds to PLCβ, thereby reducing its activity.

Next, we used embryonic fibroblasts from mice deficient in both Gα_q_ and the related Gα_11_ [38] to study structural requirements of Gα_q/11_ for inducing SphK1 translocation. In Gα_q/11_ double-deficient MEFs, both hSphK1-cerulean and YFP-mSphK1 were localized in the cytosol, and their localization did not change upon stimulation of the B_2_ bradykinin receptor unless Gα_q_-YFP or Gα_q_ wild-type were co-transfected (Figure 4). Of note, in cells expressing Gα_q_ wild-type or Gα_q_-YFP, SphK1 was localized to a small part at the plasma membrane, even under control conditions (Figure 4B,E). This was not the case in cells lacking both Gα_q_ and Gα_11_ (Figure 4A,D). Using these cells, we confirmed that stimulation of the G_i_-coupled S1P_1_ receptor did not induce SphK1 translocation, even when S1P_1_, GFP-hSphK1, and Gα_i2_ were co-transfected (Figure 4C). However, expression of the chimeric G-protein, Gα_qi5_-G66D, which links G_i_-coupled receptors to G_q_ signaling pathways [27], enabled S1P_1_ to induce SphK1 translocation (Figure 4C). In cells co-expressing the B_2_ receptor, YFP-mSphK1, and Gα_q_ wild-type, 10 µM bradykinin induced translocation of SphK1 with an average half-time of 5.8 ± 0.6 s (mean ± SEM, *n* = 31 cells; Figure 4E,G). Several Gα_q_ mutants were able to fully restore B_2_ receptor-induced SphK1 translocation in Gα_q/11_ double-deficient MEFs, with translocation half-times that did not significantly differ from that of Gα_q_ wild-type. These were Gα_q_-Y261N (t_1/2_ = 7.9 ± 1.4 s, *n* = 17), Gα_q_-D321A (t_1/2_ = 4.1 ± 0.5 s, *n* = 14), and Gα_q_-Y356K (t_1/2_ = 4.5 ± 0.5 s, *n* = 17) (all means ± SEM; Figure 4G). In cells expressing Gα_q_-W263D, B_2_ receptor-stimulated SphK1 translocation was significantly delayed, with a half-time of 10.8 ± 1.1 s (mean ± SEM, *n* = 18 cells; Figure 4G). Importantly, SphK1 translocation was very slow in cells expressing Gα_q_-T257E and typically visible only after 2–3 min, for which reason the average translocation half-time was not determined (Figure 4F). Taken together, the amino acids T257 and W263 of Gα_q_ are important for targeting of SphK1. Interestingly, Gα_q_-T257, Gα_q_-Y261, and Gα_q_-W263 are implicated in Gα_q_/GRK2 interaction, as mutation of these residues abolished Gα_q_ binding to GRK2 [31,48]. Furthermore, PLCβ activation was completely inhibited by mutation of Gα_q_-R256/T257 to alanines [49]. Finally, mutants Gα_q_-Y261N and Gα_q_-W263D had reduced binding to p63RhoGEF, while Gα_q_-T257E neither bound nor activated p63RhoGEF [33]. Thus, amino acid T257 of Gα_q_ plays a major role in binding or activation of PLCβ, p63RhoGEF, and GRK2, and likewise is important for SphK1 translocation. This result is in agreement with both hypotheses: (1) that SphK1 is activated downstream of PLCβ, and (2) that SphK1 competes with PLCβ, p63RhoGEF, and GRK2 for the same Gα_q_ binding site.

Next, we wondered which structural elements in SphK1 were required for Gα_q_-mediated translocation of the enzyme. As described above, SphK1 membrane translocation by PMA and oncogenic Ras involved CIB1, which binds to the calmodulin binding site of SphK1 [16,50]. Amino acids L194, F197, and L198 of hSphK1 were important for CIB1 binding, and the double mutant hSphK1-F197A-L198Q had reduced CIB1 binding and did not translocate to the plasma membrane in response to PMA, while its catalytic activity remained nearly intact [16,50]. Another study localized L194, F197, and L198 within a hydrophobic patch on the surface of SphK1 and demonstrated that hSphK1-L194Q and hSphK1-F197A-L198Q did not bind to acidic liposomes in vitro and were not recruited to tubular membrane invaginations induced by cholesterol extraction in living cells [4]. Hence, this hydrophobic patch is regarded as essential for curvature-sensitive membrane binding of SphK1 [9,21]. We show here that the two hSphK1 mutants, hSphK1-L194Q-GFP and hSphK1-F197A-L198Q-GFP, did not visibly translocate to the plasma membrane in response to M_3_ receptor activation in HEK-293 cells (Figure 5A–C). Interestingly, while usually there were only low levels of wild-type SphK1 in the nuclei of transfected HEK-293 cells, there was significant fluorescence in the nuclei of cells transfected with hSphK1-L194Q-GFP (Figure 5B). Furthermore, the mutant, hSphK1-F197A-L198Q-GFP, was strongly localized to the nuclei, with some cells expressing even more fluorescence in the nucleus than in the cytoplasm (Figure 5C). This observation suggests that mutations in this region might possibly disrupt a nuclear export sequence, although the two known nuclear export sequences in hSphK1 comprise amino acids 147–155 and 161–169 [51]. Because of the mentioned involvement of hSphK1-L194, -F197, and -L198 in CIB1 binding [16], we furthermore studied G_q/11_-dependent SphK1 translocation in CIB1-deficient MEFs. As shown in Figure 5D, the B_2_ receptor was clearly able to induce GFP-hSphK1 translocation in cells lacking CIB1. In addition, the translocation half-time was not altered (data not shown). Taken together, we demonstrate that this region comprising L194, F197, and L198 in hSphK1 is important for G_q/11_-dependent SphK1 translocation, very likely because this hydrophobic patch is required for membrane binding. CIB1, however, does not play a role in G_q_-mediated SphK1 translocation.

PLCβ and p63RhoGEF bind to the effector binding site of Gα_q_ primarily via their conserved ALXXPI motifs (X represents any amino acid) [40]. Although both enzymes have additional domains that contribute to the interaction with active Gα_q_, mutation of the conserved leucine in this motif (L859 in human PLCβ_3_, L475 in human p63RhoGEF) is sufficient to eliminate Gα_q_ binding [40]. Similarly, although GRK2 lacks the ALXXPI motif, it contains a structurally equivalent leucine (L118) that is essential for Gα_q_ binding [40]. Interestingly, there is a similar motif, AIXXPL, in vertebrate SphK1, with isoleucine instead of leucine in position 2 and leucine instead of isoleucine in position 6 of this motif (Figure 6A). Comparison of different SphK1 homologues shows the conservation of this motif among vertebrates, with small variations concerning the leucine/isoleucine substitutions, such as ALXXPL in *Gallus gallus* and AIXXPI in *Xenopus laevis* (Figure 6A). We did not find such a motif in non-vertebrate SphK, such as *Drosophila melanogaster* or *Caenorhabditis elegans* SphK, in agreement with the current view that S1P-GPCRs, which are ultimately targeted by SphK1 plasma membrane translocation, have evolved with the vertebrates (see [1]). To study the functional importance of the AIXXPL motif, we generated the mutants hSphK1-A100E-I101E and hSphK1-P104A-L105A as fusion proteins with *N*-terminal mCherry. When expressed in HEK-293 cells, both mCherry-hSphK1-A100E-I101E and mCherry-hSphK1-P104A-L105A were detected by an anti-mCherry antibody and by an antibody directed against the C-terminus of human SphK1, and had the same molecular weight as mCherry-hSphK1 wild-type (Figure 6B). The double bands seen in Figure 6 were also seen with mCherry alone, and thus caused by the fluorescent tag (Figure 6B). Furthermore, expression of all the SphK1 mutants elevated intracellular S1P concentrations, as measured by high-performance liquid chromatography tandem mass spectrometry. In cells transfected with mCherry, the concentration of S1P was 1050 ± 150 pg/mg protein (*n* = 8), while expression of mCherry-hSphK1 wild-type increased S1P to 1500 ± 200 pg/mg protein (*n* = 6; *p* < 0.001). Cells expressing mCherry-hSphK1-A100E-I101E had S1P concentrations of 1300 ± 160 pg/mg protein (*n* = 9; *p* < 0.05), and cells expressing mCherry-hSphK1-P104A-L105A had 1500 ± 190 pg/mg protein (*n* = 9; *p* < 0.001) (all values represent means ± SD, with significance tested in one-way ANOVA). These results indicated that all of the mutants were catalytically active.

Next, we studied plasma membrane translocation of the two mutants in response to M_3_ receptor activation in HEK-293 cells. In unstimulated cells, both mCherry-hSphK1-A100E-I101E and mCherry-hSphK1-P104A-L105A were localized in the cytosol of the cells and only a small part was sometimes seen in the nucleus, similar to mCherry-SphK1 wild-type (Figure 6C–E). Interestingly, carbachol-induced translocation of mCherry-hSphK1-A100E-I101E occurred in only 6 of 24 cells (25%), while mCherry-hSphK1 wild-type translocated in 25 of 26 cells (96%) in this set of experiments (Figure 6F). The translocation half-times of mCherry-hSphK1-A100E-I101E in the 6 cells with translocation were not significantly different from those of mCherry-hSphK1 wild-type (Figure 6F). However, translocation efficiency, measured as the % decrease in cytosolic fluorescence, was significantly lower with mCherry-hSphK1-A100E-I101E than with the wild-type enzyme (14.0 ± 2.5%, *n* = 25 versus 23.6 ± 1.4%, *n* = 6; mean ± SEM; Figure 6D,F). The other mutant, mCherry-hSphK1-P104A-L105A, translocated in 19 of 34 cells stimulated with carbachol (56%), while mCherry-hSphK1 wild-type translocated in 100% of cells in this set of experiments (Figure 6E,F). Again, in the cells which had a response, translocation efficiency of this mutant was significantly reduced (16.5 ± 1.6%, *n* = 19 versus 23.4 ± 1.5%, *n* = 16; mean ± SEM), while translocation half-times were not significantly different when compared to the wild-type enzyme (Figure 6F). In cells co-transfected with constitutively active Gα_q_-Q209L-EE, mCherry-hSphK1-A100E-I101E remained cytosolic in the majority of cells, while mCherry-hSphK1-P104A-L105A was localized at the plasma membrane to a variable extent (Figure 6G). Quantification of the plasma membrane/cytosol fluorescence ratios revealed that Gα_q_-Q209L-EE-induced membrane attachment of both mutants was significantly reduced compared to the wild-type enzyme, and that the A100E-I101E mutant was again more affected (Figure 6H). Thus, the results for the two mutants support a role for the AIXXPL motif in G_q_ targeting of SphK1. G_q_-mediated translocation was much more strongly affected by mutation of A100 and I101 to glutamate than by mutation of P104 and L105 to alanine. This might be explained by the stronger disruption of the domain by the negative charges of the two glutamates, or by a higher relevance of AI compared to PL. Intriguingly, within the AIXXPL motif, I101 corresponds to L859 in human PLCβ_3_ and L475 in human p63RhoGEF, which are most important for G_q_ binding [40]. We think that the mutations did not disrupt the general structure of hSphK1, as (1) the molecular weight and subcellular localization in unstimulated cells were normal, (2) both mutants were able to translocate at least in some cells, and (3) S1P concentrations were elevated in cells expressing SphK1-A100E-I101E and SphK1-P104A-L104A indicating catalytic activity. The remaining responses of the mutants might be due to incomplete disruption of the domain, contribution of other parts of the enzyme (see below), or to a second parallel pathway that might involve phosphorylation.

Taken together, we present functional data showing that SphK1 translocation by G_q/11_-coupled receptors is prevented by (over)expression of diverse Gα_q/11_ effectors or binding partners, suggesting that SphK1 is targeted either via PLCβ or directly by activated Gα_q/11_. The fact that expression of catalytically inactive hSphK1-G82D reduced receptor-stimulated inositol phosphate production argues in favor of the latter possibility, although the role of PLCβ in this scenario remains unclear. Furthermore, we show that SphK1′s conserved AIXXPL motif is involved in translocation of the enzyme by Gα_q_. Further studies are required to examine whether this motif indeed mediates direct interaction of SphK1 with Gα_q/11_. We do not exclude that there are other structural elements in SphK1 that directly or indirectly interact with Gα_q_. In fact, we had shown before that both the *N*-terminus and C-terminus of hSphK1 (except for the TRAF2 binding site) were required for M_3_ receptor-induced translocation [23]. The fragment with *N*-terminal deletion of the first 110 amino acids (hSphK1^111−384^), thus without AIXXPL motif, did not translocate. The fragment with C-terminal deletion of 27 amino acids in addition to the TRAF2 binding site (hSphK1^1−350^) also did not translocate, suggesting that there are other unknown elements in this region required for interaction with Gα_q/11_, the membrane, or other regulatory proteins (see also discussion in [21]).

There are numerous examples of the importance of SphK1 in G_q/11_ signaling and functional responses. G_q/11_-coupled receptors engaging SphK1 include not only muscarinic receptors [52] and bradykinin receptors [25], but also protease-activated receptors [53,54], angiotensin receptors [55], or histamine receptors [56,57], just to name a few. These examples show that SphK1 was involved in G_q/11_-dependent regulation of the vascular endothelium and smooth muscle, G_q/11_-mediated myogenic differentiation of skeletal muscle, or G_q/11_-regulated inflammatory responses. From its eminent role in vascular regulation, it was hypothesized recently that SphK1 might be a therapeutic target in pulmonary hypertension [58], in addition to its roles in inflammation, fibrosis, and cancer [10]. Constitutively active G_q/11_ proteins act as oncogenes [59], and thus given the role of SphK1 in cancer, it will be interesting to unravel a possible interconnection. Another important theme is the emerging role of SphK1 in epithelial–mesenchymal transition [60,61]; however, it should be kept in mind that besides G_q/11_ proteins, there are many other pathways regulating SphK1 expression and activity, and which may be important in this context. Nevertheless, we strongly believe that unravelling the mechanism(s) by which G_q/11_ regulates SphK1 will further help to understand the functional roles of this enzyme and facilitate its targeting by potential therapeutics.

## Figures and Tables

**Figure 1 cells-09-02201-f001:**
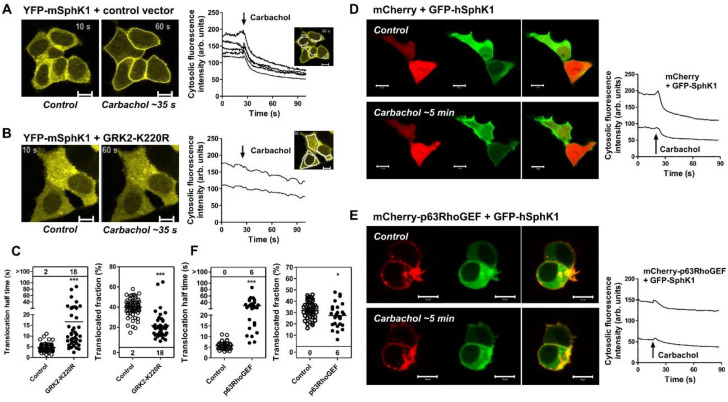
Influence of catalytically inactive GRK2 and p63RhoGEF on M_3_ receptor-induced SphK1 translocation. HEK-293 cells stably expressing the M_3_ muscarinic acetylcholine receptor were transfected as indicated and translocation of SphK1 was monitored by live cell imaging with a confocal laser scanning microscope. (**A**–**C**) Cells were transfected with YFP-mSphK1 and either catalytically inactive GRK2 (GRK2-K220R) or control vector. Time series were acquired with ~3 images/s and 100 µM carbachol was added after ~25 s. Images were taken from representative time series at 10 and 60 s, thus showing localization of YFP-mSphK1 before and ~35 s after stimulation with carbachol, respectively. The line graphs show the corresponding time courses of cytosolic fluorescence intensity, measured in the indicated cytosolic regions. (**C**) The SphK1 translocation half-times were measured by fitting exponential curves to the translocation-induced decay in cytosolic fluorescence intensity. Each dot represents a single cell. Translocation half-times >100 depict the number of cells which did not respond. Note: *** *p* < 0.0001 in *t*-tests with Welch’s correction for unequal variances. (**D**–**F**) Cells were transfected with GFP-hSphK1 (green) and either mCherry or mCherry-p63RhoGEF (red). Representative images were taken at high spatial resolution immediately before and after the acquisition of time series, during which only GFP fluorescence was monitored with 1 image/s. The line graphs showing time courses of cytosolic fluorescence intensity correspond to the cells shown in the images. Carbachol was added at the indicated time points. (**F**) SphK1 translocation half-time was measured as described in (C). Note: * *p* < 0.05, *** *p* < 0.0001 in *t*-tests with Welch’s correction for unequal variances. Micrometer bars, 10 µm.

**Figure 2 cells-09-02201-f002:**
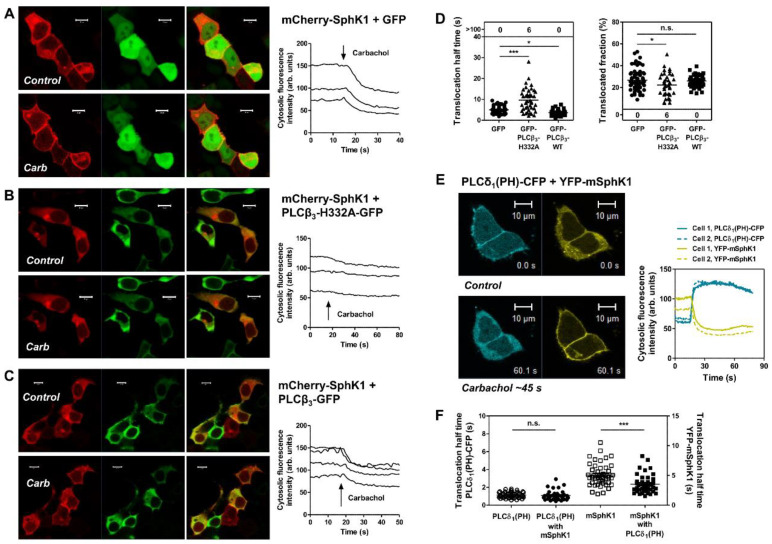
Influence of PLCβ_3_ and PLCδ(PH) on M_3_ receptor-induced SphK1 translocation. HEK-293 cells stably expressing the M_3_ receptor were transfected as indicated. (**A**–**D**) Cells were transfected with mCherry-hSphK1 (red) and GFP (**A**), catalytically inactive PLCβ_3_ (PLCβ_3_-H332A-GFP) (**B**) or PLCβ_3_ wild-type (PLCβ_3_-WT) (**C**). Representative images were taken at high spatial resolution immediately before and after the acquisition of time series during which only mCherry fluorescence was monitored at 1 image/s. The line graphs showing time courses of cytosolic fluorescence intensity correspond to the cells shown in the images. Carbachol was added at the indicated time points. (**D**) SphK1 translocation half-times were measured by fitting exponential curves to the decay in cytosolic fluorescence. The translocated fraction represents the decay in cytosolic fluorescence intensity in % of initial fluorescence. Each dot represents a single cell. Translocation half-times >100 depict the number of cells which did not respond. Note: n.s., not significant; * *p* <0.05, *** *p* < 0.0001 in *t*-tests with Welch’s correction for unequal variances. (**E**,**F**) Cells were transfected with PLCδ(PH)-CFP (cyan), YFP-mSphK1 (yellow), or both. Carbachol-induced translocation of the two proteins was studied in time series with ~3–4 images/s. (**E**) Images were taken from a representative time series with double-transfected cells before and ~45 s after addition of carbachol. The line graph shows the time course of cytosolic fluorescence intensity for both CFP and YFP. (**F**) Translocation half-times for PLCδ(PH)-CFP and YFP-mSphK1, both alone and in combination. Note: n.s., not significant; *** *p* < 0.0001 in *t*-tests with Welch’s correction for unequal variances. Micrometer bars, 10 µm.

**Figure 3 cells-09-02201-f003:**
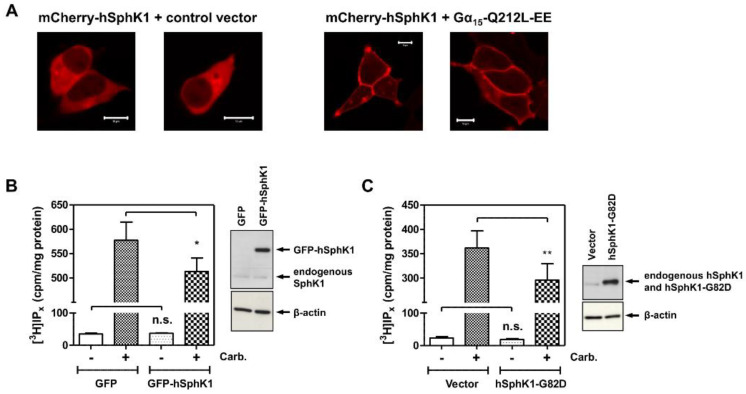
(**A**) SphK1 translocation by constitutively active Gα_15_. HEK-293 cells stably expressing the M_3_ receptor were transfected with mCherry-hSphK1 and either control vector or Gα_15_-Q212L-EE. Shown are two representative images each. Bars, 10 µm. (**B**,**C**) Influence of SphK1 overexpression on M_3_ receptor-induced inositol phosphate production. HEK-293 cells were transfected with GFP or GFP-hSphK1 (**B**), or with the pcDNA3.1 vector or hSphK1-G82D (**C**). The formation of [^3^H]inositol phosphates ([^3^H]IP_x_) was measured in [^3^H]inositol-labelled cells stimulated with 100 µM carbachol (Carb.) for 20 min in the presence of LiCl. The expression of GFP-hSphK1 and hSphK1-G82D was confirmed with an anti-SphK1 antibody. The data are means ± SEM from *n* = 9 (B) or *n* = 8 (C) independent experiments, each performed in triplicate. Note: n.s., not significant; * *p* < 0.05, ** *p* = 0.01 in paired *t*-test.

**Figure 4 cells-09-02201-f004:**
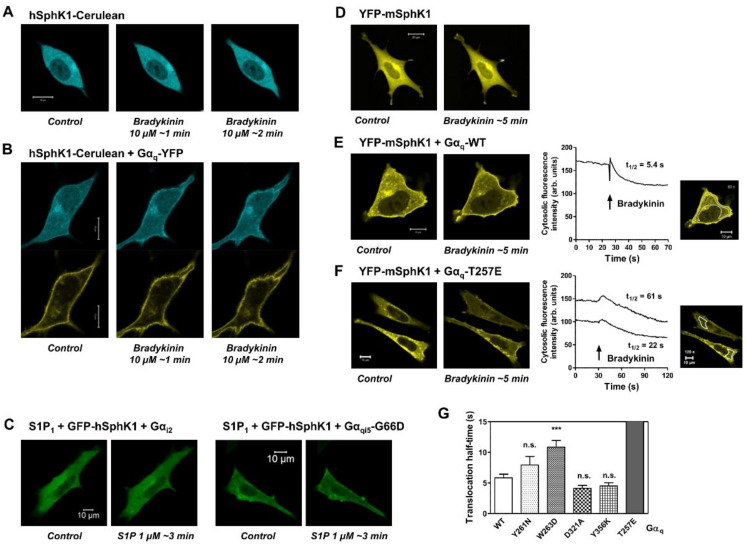
Identification of Gα_q_ residues required for G_q/11_-mediated SphK1 translocation. (**A**–**G**) SphK1 translocation was analyzed in Gα_q/11_-double-deficient MEFs. (**A**,**B**) The cells were co-transfected with the bradykinin B_2_ receptor and hSphK1-cerulean without (**A**) and with (**B**) Gα_q_-YFP. Images were taken before and after addition of 10 µM bradykinin as indicated. (**C**) The cells were co-transfected with the S1P_1_ receptor, GFP-hSphK1, and either Gα_i2_ or Gα_qi5_-G66D as indicated. Images were taken before and ~3 min after addition of 1 µM S1P. (**D**–**G**) The cells were co-transfected with the B_2_ receptor, YFP-mSphK1, and Gα_q_ wild-type (Gα_q_-WT) or various Gα_q_ mutants as indicated. Images were taken at a high resolution before and ~5 min after addition of 10 µM bradykinin. The time course of SphK1 translocation was measured by taking series of images at lower spatial resolution at ~1 image/400 ms. Cytosolic fluorescence intensity was measured in selected regions as indicated in the inserts in (**E**,**F**), and translocation half-times were calculated by fitting exponential curves to the decay in the cytosolic fluorescence intensity. (**G**) Translocation half-times obtained with the various Gα_q_ mutants. Data are means ± SEM; *n* = 31 (Gα_q_-WT); *n* = 14–18 (Gα_q_ mutants). Note: n.s., not significant; *** *p* < 0.0001 in one-way ANOVA followed by Bonferroni´s post-test. The micrometer bars represent 10 µm.

**Figure 5 cells-09-02201-f005:**
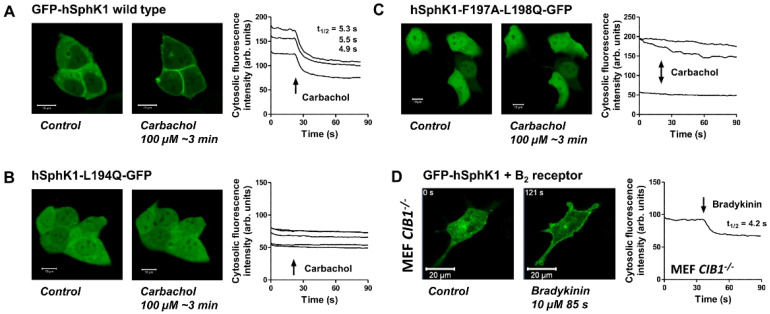
Role of the hydrophobic patch in SphK1 for G_q/11_-mediated SphK1 translocation. (**A**–**C**) HEK-293 cells stably expressing the M_3_ receptor were transfected with GFP-hSphK1, hSphK1-L194Q-GFP, or hSphK1-F197A-L198Q-GFP. Translocation of SphK1 mutants was studied upon stimulation of the cells with 100 µM carbachol. Shown are images before and after stimulation, and time courses of cytosolic fluorescence from representative experiments. In (**C**), the cytosolic fluorescence of the cell in the upper left could not be evaluated because of the strong change in cell shape. The micrometer bars represent 10 µm. (**D**) Role of CIB1 for G_q_-mediated SphK1 translocation. MEFs from CIB1-deficient mice were transfected with GFP-hSphK1 and the B_2_ receptor. Shown are images and time courses of cytosolic fluorescence from a representative time series during which 10 µM bradykinin was added after 35 s. The two images, thus, show localization of GFP-hSphK1 before and 85 s after addition of bradykinin. Micrometer bars, 20 µm.

**Figure 6 cells-09-02201-f006:**
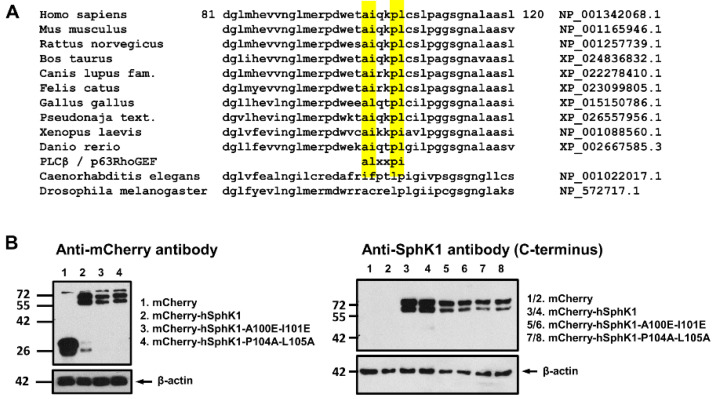
Identification of structural elements in SphK1 required for G_q/11_-mediated translocation. (**A**) Sequence alignment of diverse SphK1 homologues between amino acids 81–120 of human SphK1. (**B**) Western blot analysis of mCherry-hSphK1-A100E-I101E and mCherry-hSphK1-P104A-L105A expressed in HEK-293 cells. (**C**–**E**) Analysis of localization and translocation of mCherry-hSphK1-A100E-I101E and mCherry-hSphK1-P104A-L105A in HEK-293 cells stably expressing the M_3_ receptor. The cells were stimulated with 100 µM carbachol as indicated. Shown are images before and after stimulation, and time courses of cytosolic fluorescence from representative experiments. Since translocation of mCherry-hSphK1-P104A-L105A was variable, two examples are shown here for this mutant. (**F**) Summary of translocation half-times and translocated fractions from (C–E). Each dot represents a single cell. Translocation half-times >100 depict the number of cells which did not respond. Note: ** *p* < 0.01 in unpaired *t*-test. (**G**,**H**) Influence of constitutively active Gα_q_ on subcellular localization of hSphK1 mutants. HEK-293 cells were transfected with mCherry-hSphK1 wild-type, mCherry-hSphK1-A100E-I101E, or mCherry-hSphK1-P104A-L105A, plus either control vector or Gα_q_-Q209L-EE as indicated. (**G**) Representative images. All micrometer bars, 10 µm. (**H**) Quantification of SphK1 plasma membrane localization was performed by measuring the fluorescence profiles of individual cells and calculating the plasma membrane/cytosol fluorescence ratios. Each dot represents a single cell. Note: * *p* < 0.05, *** *p* < 0.0001 in one-way ANOVA followed by Dunnett’s multiple comparisons test.

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
