# Peer review of "Dissecting Gq/11-Mediated Plasma Membrane Translocation of Sphingosine Kinase-1"

_cells, 2020, doi:10.3390/cells9102201_

Round 1

Reviewer 1 Report

Sphingosine kinase 1, SphK1, belongs to the lipid kinase family, which specifically phosphorylates sphingosine to produce sphingosine 1-phosphate, S1P – the multifunctional bioactive lipid mediator. Regulation of expression and activity of SphK1 is complex and combines transcriptional, translational and post-translational mechanisms.

In the current study the authors studied regulation of SphK1 by G-coupled receptors with a major focus given to Gq/11 signaling pathways that trigger a rapid and long-lasting SphK1 translocation from cytosol to the plasma membrane. The authors used a set of plasmids with diverse fluorescent tags for overexpression of the murine and human wild-type variants of SphK1 as well as the numerous mutant forms. The experiments were done in in vitro cell-based systems using HEK-293 cells and mouse embryonic fibroblasts from mice deficient in particular G proteins/signaling. SphK1 translocation was analyzed by confocal laser scanning microscopy using as estimation parameter the translocation half times on the basis of the fluorescence intensity within defined cytosolic regions. The authors also confirmed the catalytic activity of mutated SphK1 constructs and showed that overexpression resulted in elevated intracellular S1P levels by the factor of 1.3 – 1.5. Cellular S1P was determined by high-performance liquid chromatography tandem mass spectrometry.

The authors present functional data showing that SphK1 translocation by Gq/11-coupled receptors is prevented by (over)expression of diverse Gαq/11 effectors or binding partners, suggesting that SphK1 is targeted either via PLCβ or directly by activated Gαq/11. The authors conclude that the fact that expression of catalytically inactive hSphK1-G82D reduced receptor-stimulated inositol phosphate production argues in favor of the latter possibility, although the role of PLCβ in this scenario remains unclear. Furthermore, the authors identified the SphK1’s conserved AIXXPL motif that is involved in translocation of the enzyme by Gαq.  

This comprehensive study adds novel layer of information to our understanding of complex pattern of SphK1 regulation.  

Minor comments:

Line 180: font differences to other text

As stated by authors, there are numerous examples for the importance of SphK1 in signaling and functional responses.  There are indications that the upregulation of SPHK1 and the subsequent increase in intracellular levels of S1P might be involved in the epithelial to mesenchymal transition program, EMT. This aspect, however, has been much less studied and less understood (few published examples Meshcheryakova A et al. Oncotarget. 2016;7(16):22295-22323. doi:10.18632/oncotarget.7947; Liu SQ et al.  Int J Oncol. 2019 Jan;54(1):41-52. doi: 10.3892/ijo.2018.4607). The authors may wish to discuss/comment on the potential relevance of their findings for EMT - the physiologically and pathologically important process.   

Author Response

Answer to Reviewer No 1

First of all, we thank the reviewer for his/her careful evaluation of our manuscript. We would like to answer as follows:

“Line 180: font differences to other text“

Thank you, we have corrected this.

“There are indications that the upregulation of SPHK1 and the subsequent increase in intracellular levels of S1P might be involved in the epithelial to mesenchymal transition program, EMT. This aspect, however, has been much less studied and less understood (few published examples Meshcheryakova A et al. Oncotarget. 2016;7(16):22295-22323. doi:10.18632/oncotarget.7947; Liu SQ et al. Int J Oncol. 2019 Jan;54(1):41-52. doi: 10.3892/ijo.2018.4607). The authors may wish to discuss/comment on the potential relevance of their findings for EMT - the physiologically and pathologically important process.”

The role of the SphK1/S1P pathway in EMT is indeed a highly interesting emerging theme. However, SphK1 expression and activity can be regulated by many pathways, of which Gq/11 proteins are just one. Since to our knowledge the role of Gq/11 proteins in EMT is not really clear, we have only briefly mentioned this in the last section of “Results and Discussion”.

Reviewer 2 Report

In this study Blankenbach et al. investigated the mechanisms underlying agonist-and Galpha-induced sphingosine kinase (SphK1) translocation using membrane translocation assays of fluorescently tagged SphK1 variants. They showed that overexpressing a number of known Galphaq effectors interfered with agonist-induced SphK1 translocation. They identified a putative effector domain mutation of Galphaq (T257) as being important for SphK1 translocation. Finally, they identified a motif within SphK1 that is shared by other Galphaq effectors and that is distinct from previously identified membrane association domains of SphK1 that promotes agonist- and Galphaq-induced translocation of SphK1 to the plasma membrane without altering SphK1 catalytic activity.  They concluded that the interaction of SphK1 with activated Galphaq may be an important mechanism by which agonists induce Sphk1 membrane translocation. The study provides a possible novel insight into the mechanisms by which SphK1 is activated in vivo that could explain how agonists promote SphK1 translocation to the plasma membrane. There are two major weaknesses in the study. First, although the data is consistent with a direct interaction between Sphk1 and Galphaq, the paper presents no binding data, which would be feasible and essential to support the conclusion. Second, the conclusion that the AIXXPL motif is important for SPhK1 membrane translocation is based on data that seems preliminary and not well quantified, raising concerns about the justification of the conclusion.

Major concerns:

  1. The authors present numerous pieces of indirect evidence suggesting that the binding of SphK1 with activated Galphaq may be critical for SphK1 translocation. Indeed, the authors have established the importance of a Galphaq in SphK1 translocation in previous studies. Here they identify a known effector residue in Galphaq required for SPHK1 translocation, and they identify a motif shared by Galphaq effectors that is required for SPhK1 translocation. Yet they state in line 463 that further studies are needed to demonstrate direct interaction between Galphaq and SphK1. The paper would be greatly strengthened if it included binding studies that address whether there is indeed an interaction between Galphaq/11 and SphK1.
  2. Figure 6. The interesting and potentially novel part of this study is the identification of the AIXXPL motif that the authors conclude is important for G protein-mediated SPHK-1 membrane translocation. However, the interpretation of data in Figure 6 is problematic. First, there seems to be a significant decrease in 104A105A translocation in response to carbachol (Fig 6F) but not following activation of Galphaq (Fig 6H). How to the authors reconcile this apparent discrepancy? Considering this, is the conclusion stated on line 443 justified? Second, the 100E101E mutations do seem to reduce both carbachol- and activated Galphaq-mediated SphK1 translocation (Figs 6F and 6H). However, in Fig 6F, only six 100E101E expressing cells were examined. This seems to be massively under sampled and given the variability of the translocation assay, could lead to erroneous conclusions. In addition, in the data examining activated Galphaq in Fig 6E was not quantitative since no statistics were used to determine whether 100E101E mutants were significantly different from wild type controls and the criteria for determining “strong TL”, and “weak TL” were not described and seem inherently semi-quantitative at best. Finally, the variability of the membrane translocation assay even for wild type SphK1 (See Fig 6F and 6H) raises the possibility that the expression levels of the SphK1 transgenes may influence its membrane recruitment. Indeed, the 100E101E mutant seems to be expressed at lower levels than wild type by western blot (Fig 6B- although it is hard to tell since quantification of expression levels was not done) and generates less SIP than wild type (line 422) raising the concern that the reduced membrane translocation of this mutant may be an indirect result of lower expression levels. These concerns need to be expressed experimentally before the conclusion that the AIXXPL motif is important for SPHK-1 translocation can be justified.
  3. All Figures. Examining SPHK1 translocation to the plasma membrane by measuring cytoplasmic fluorescence seems indirect. Is there a way of examining changes in plasma membrane fluorescence directly and perhaps using changes in both cytosolic and plasma membrane fluorescence intensities as complementary measurements for PM association?

Author Response

Answer to Reviewer No 2

We thank the Reviewer for his/her careful evaluation of our manuscript. Please find below our point-by-point answer to the comments and questions.

  1. “The paper would be greatly strengthened if it included binding studies that address whether there is indeed an interaction between Galphaq/11 and SphK1.”

We fully agree. In fact, we already have performed some binding studies but noticed that comprehensive and reliable Gq/SphK1 interaction studies will probably require a minimum of one more year of additional work. Nevertheless, we think that our functional data including identification of a domain in SphK1 relevant for targeting by Gq are interesting for readers and other researchers working in the field. We strongly hope that we can convince the Reviewer that publication of our functional data is of sufficient interest to the public.

  1. “Figure 6. The interesting and potentially novel part of this study is the identification of the AIXXPL motif that the authors conclude is important for G protein-mediated SPHK-1 membrane translocation. However, the interpretation of data in Figure 6 is problematic. First, there seems to be a significant decrease in 104A105A translocation in response to carbachol (Fig 6F) but not following activation of Galphaq (Fig 6H). How to the authors reconcile this apparent discrepancy? Considering this, is the conclusion stated on line 443 justified? “

Fig. 6H shows our semi-quantitative evaluation of the number of cells with no, weak and strong translocation, induced by constitutively active Gαq in the three groups of SphK1 wild type-, SphK1-A100E-I101E-, or SphK1-P104A-L105A-expressing cells. We agree with the Reviewer (regarding also the comment further down) that the evaluation is semi-quantitative – at least it was performed in a blinded manner by a person not involved in these experiments. Because of this limitation, we did not put the statistical evaluation into the manuscript. In fact, Gαq-Q209L-induced translocation of both SphK1-A100E-I101E (p<0.0001) and SphK1-P104A-L105A (p<0.05) was significantly different from SphK1 wild type in Χ2 test (using the numbers of cells in each group, not the percentage). Therefore, we think that our statement on line 443, “the P104A-L105A mutant translocated in cells co-transfected with Gαq-Q209L-EE, albeit not always as effectively as the wild type enzyme“ is indeed supported by the data.

“Second, the 100E101E mutations do seem to reduce both carbachol- and activated Galphaq-mediated SphK1 translocation (Figs 6F and 6H). However, in Fig 6F, only six 100E101E expressing cells were examined. This seems to be massively under sampled and given the variability of the translocation assay, could lead to erroneous conclusions.”

We are sorry that the data were presented in a misleading manner. Fig. 6F shows altogether 24 cells expressing the A100E-I101E mutant. While there were indeed only six cells in which the A100E-I101E mutant translocated, there were18 cells in which the mutant did not translocate at all, indicated as “translocation half time >100 s”. Same with the P104A-L105A mutant: it did not translocate in 15 cells. We have now tried to make this more clear in the Figure Legends. It might also be a problem that the Figures were set to a quite small size.

“In addition, in the data examining activated Gαq in Fig 6E was not quantitative since no statistics were used to determine whether 100E101E mutants were significantly different from wild type controls and the criteria for determining “strong TL”, and “weak TL” were not described and seem inherently semi-quantitative at best. “

This comment concerns Figure 6H. We agree with the Reviewer that the evaluation is semi-quantitative. At least it was performed in a blinded manner by a person not involved in these experiments (this is now mentioned in “Data analysis”). Statistical analysis by Χ2 test showed that both SphK1-A100E-I101E and SphK1-P104A-L105A behaved significantly differently from SphK1 wild type (see above). Since we agree that the data is just semi-quantitative, we did not mention the statistics in the manuscript. The Figure is intended to illustrate that Gαq-Q209L-induced translocation of SphK1 is much stronger affected by the EE- than by the AA-mutation. We have now explicitly pointed out in the Figure Legend that the evaluation was semi-quantitative. Please let us know whether we should better omit this Figure, or include the statistical evaluation into the manuscript.

“Finally, the variability of the membrane translocation assay even for wild type SphK1 (See Fig 6F and 6H) raises the possibility that the expression levels of the SphK1 transgenes may influence its membrane recruitment. Indeed, the 100E101E mutant seems to be expressed at lower levels than wild type by western blot (Fig 6B- although it is hard to tell since quantification of expression levels was not done) and generates less SIP than wild type (line 422) raising the concern that the reduced membrane translocation of this mutant may be an indirect result of lower expression levels. These concerns need to be expressed experimentally before the conclusion that the AIXXPL motif is important for SPHK-1 translocation can be justified.”

Figures 6F and 6H show different conditions: Figure 6F shows the dynamic response to carbachol, measured as stimulation-induced decay in cytosolic fluorescence intensity, while Fig. 6H shows the response in cells expressing constitutively active Gαq, which is a static response. Comparison of the dynamic responses of wild type SphK1 to carbachol (Figures 1C, 1F, 2D, 4G, 6F) shows that this response was highly reproducible, even in different cell types (HEK-293 and MEF) and with different receptors (M3 and B2).

Importantly, the expression levels of SphK1 shown in the Western blots are not relevant for the studies in single cells, because we preferentially selected cells that expressed the fluorescent SphK1 constructs at an intermediate level. In the few cases in which there was a cell with weak expression of SphK1, we did not notice a different translocation half time. For example, Fig. 6C shows 3 cells with intermediate expression, 1 cell with a slightly higher expression (upper left) and 1 cell with lower expression of SphK1 (upper right). The cytosolic fluorescence intensity at baseline (Y0, reflecting the relative expression levels) and translocation half times were as follows: cell 1: Y0=129, t1/2=5.5 s; cell 2: Y0=194, t1/2=6.8 s; cell 3: Y0=111, t1/2=5.5 s; cell 4: Y0=106, t1/2=1.9 s; cell 5: Y0=58, t1/2=5.3 s. Thus, cell No 5 with a relatively low expression level had a translocation half time of 5.3 s which is an average value. Similar observations were made in many more cells.

  1. “All Figures. Examining SPHK1 translocation to the plasma membrane by measuring cytoplasmic fluorescence seems indirect. Is there a way of examining changes in plasma membrane fluorescence directly and perhaps using changes in both cytosolic and plasma membrane fluorescence intensities as complementary measurements for PM association?”

It is indeed difficult to measure the increase in PM fluorescence upon translocation in our system because of cell shape changes in response to stimulation. In particular, the PM moved and changed its shape upon stimulation with carbachol. This is the reason why we relied on measurements of cytosolic fluorescence intensities. Interestingly, translocation half-times of carbachol-stimulated SphK1 translocation were very similar over a broad range of experiments and on average always ~5 s (see Figures 1C, 1F, 2D, 4G, 6F), indicating that this is a reliable and robust method.

Taken together, we hope that we can convince the Reviewer that we present reliable functional data, which merit publication even though direct Gαq/SphK1 interaction data will remain subject of our future studies.

Round 2

Reviewer 2 Report

Each of my concerns have been adequately addressed except for those regarding Fig 6G and H. While it is appreciated that the authors added a sentence to the Methods that the data was scored blind, the did not state the criteria used to score the images as "weak TL" or "strong TL". Furthermore, TL (translocation) seems to be an inappropriate term to use here since Galpha is chronically activated in these cells so there is no control to measure translocation against in a given cell. My recommendation is either to remove 6G and H, or to state the criteria used for scoring PM association in the Methods, or to reevaluate the images with an improved scoring methodology.

Author Response

Answer to Reviewer #2:

Thank you for insisting. We chose to reevaluate the images with a quantitative methodology. Using the "profile" option of the ZEN software, we measured the fluorescence profiles of the cells and calculated the ratios of plasma membrane and cytosolic fluorescence intensities. The result is shown in the new Fig. 6H: in cells expressing Galphaq-Q209L-EE, both SphK1 mutants were significantly less localized at the plasma membrane than the wild type enzyme, and the EE mutant was stronger affected than the AA mutant.

We think that you were right and this is indeed an improvement. We hope that you agree and will finally accept our manuscript for publication.

Thank you again and kind regards,

Dagmar Meyer zu Heringdorf